# A Brief Overview of Neutrophils in Neurological Diseases

**DOI:** 10.3390/biom13050743

**Published:** 2023-04-25

**Authors:** Supriya Chakraborty, Zeynab Tabrizi, Nairuti Nikhil Bhatt, Sofia Andrea Franciosa, Oliver Bracko

**Affiliations:** 1Department of Biology, University of Miami, Coral Gables, FL 33146, USA; 2Department of Neurology, University of Miami-Miller School of Medicine, Miami, FL 33136, USA

**Keywords:** neutrophils, NETosis, vascular inflammation, neutrophil heterogeneity

## Abstract

Neutrophils are the most abundant leukocyte in circulation and are the first line of defense after an infection or injury. Neutrophils have a broad spectrum of functions, including phagocytosis of microorganisms, the release of pro-inflammatory cytokines and chemokines, oxidative burst, and the formation of neutrophil extracellular traps. Traditionally, neutrophils were thought to be most important for acute inflammatory responses, with a short half-life and a more static response to infections and injury. However, this view has changed in recent years showing neutrophil heterogeneity and dynamics, indicating a much more regulated and flexible response. Here we will discuss the role of neutrophils in aging and neurological disorders; specifically, we focus on recent data indicating the impact of neutrophils in chronic inflammatory processes and their contribution to neurological diseases. Lastly, we aim to conclude that reactive neutrophils directly contribute to increased vascular inflammation and age-related diseases.

## 1. Introduction

Neutrophils are human blood’s most abundant white blood cells making up 60–70% of total leukocytes [1]. Neutrophils originate in the bone marrow from hematopoietic stem cells, and after going through successive differentiation processes, they circulate in the blood for several hours to days [2] and transmigrate out of the blood vessels into the tissue upon encountering pro-inflammatory signals [3]. Ongoing inflammation leads to up-regulation of selectins such as P-selectin on endothelial cells, which interacts with the sialo-mucin ligand P-selectin-glycoprotein–ligand-1 (PSGL-1), constitutively expressed on the surface of neutrophils. This interaction slows down the speed of free-flowing neutrophils in the blood vessel by mediating neutrophil rolling (stop and go movement) along the endothelium [4] (Figure 1a). Following the step of neutrophil rolling, a high-affinity interaction of CD18 integrins CD11a/CD18 (LFA-1) and CD11b/CD18 (Mac-1) expressed on neutrophils with inter-cellular-adhesion-molecule-1 or -2 (ICAM-1/ICAM-2) expressed on endothelium mediates the firm arrest (complete stop) of neutrophils on the vascular endothelium [5]. Neutrophils then crawl and cross the border known as the blood–brain barrier (BBB) that lies between endothelial cells and the tissue site through a process known as diapedesis. Inflammatory signals, such as complement protein C5a, Interleukin-8, and Leukotriene B4, are among the most potent chemo-attractants that guides the neutrophil within the tissue [6,7] (Figure 1a).

Neutrophils eliminate pathogens in one of four ways. First, they can phagocytose pathogens that come into cells as phagosomes, which are fused with the lysosome and degraded through various enzymes, such as acid hydrolases [8] (Figure 1b). Second, neutrophils can release granules containing cytotoxic enzymes that destroy pathogens [9] (Figure 1b). Third, they can eject their DNA chromatin with cytotoxic granules to trap the pathogen in a process known as NETosis [10] (Figure 1b). Fourth, neutrophils also undergo oxidative bursts [11]. After the elimination of pathogens, most neutrophils undergo apoptosis or other methods of cell death, and cell debris is phagocytosed by macrophages [12]. The remaining neutrophils reverse migrate into the blood and go to the bone marrow, spleen, and liver for degradation [3]. Despite playing vital roles in the body’s defense, neutrophils are responsible for initiating or progressing various diseases such as psoriasis, rheumatoid arthritis, systemic lupus erythematosus, coronary heart disease, and asthma [13]. Moreover, the involvement of neutrophils in low-level chronic inflammatory diseases such as Alzheimer’s is being investigated [14,15].

**Figure 1 biomolecules-13-00743-f001:**
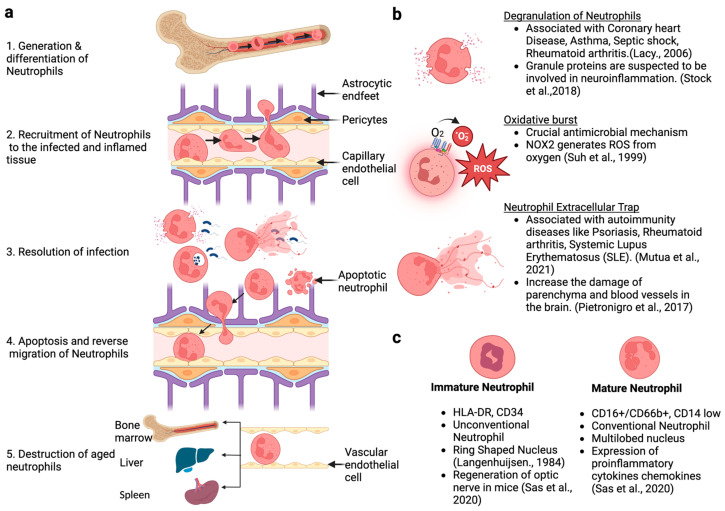
A brief overview of the neutrophil life cycle. (**a**) Generation and maturation of the neutrophil in the bone marrow and their entry into the blood after attaining maturity, functional responses upon encountering pathogens, and apoptosis or clearance after pathogen clearance; (**b**) mechanism of effector actions of neutrophil; (**c**) distinctions between immature and mature neutrophils based on cellular morphology and molecular markers [13,16,17,18,19,20,21,22,23].

With the implementation of single-cell sequencing, sophisticated programs, and software to analyze the data [24], the utilization of bone marrow chimera mice [25] is challenging the idea of a single homogeneous ‘type’ of neutrophils. Instead, it is becoming increasingly clear that the circulating neutrophil population is heterogeneous [24,26] (Figure 1c). Therefore, this review will discuss the overall development and differentiation of neutrophils, their heterogeneity, and their involvement in neurological diseases.

## 2. Generation, Development, and Life Cycle of Neutrophils

Like all leukocytes, neutrophils begin their life cycle in the bone marrow from hematopoietic stem cells, which differentiate into multipotent precursor (MPP) cells. MPPs then divide and generate lymphoid and myeloid progenitor (LMP) cells. LMPs undergo subsequent differentiation to give rise to granulocyte–macrophage precursors (GMP), which divide to generate neutrophil precursors [27,28]. Neutrophil precursors differentiate into myeloblast, promyelocyte, myelocyte, metamyelocyte, and band cell stages, which ultimately give rise to segmented neutrophils [16,17].

Myeloblasts are considered the earliest neutrophil progenitors, with a large nucleus and cytoplasm devoid of granules. Promyelocytes are larger than myeloblasts, the shape of their nucleus is rounded, and it contains azurophilic granules with myeloperoxidase as the primary compound [17,28]. In the myelocyte stage, the rounded nucleus of the promyelocyte begins to indent, and the cell size also decreases. At this stage of neutrophil development, secondary granules start forming, made up of compounds such as lactoferrin, lysozyme, and collagenase [3]. In the metamyelocyte stage, the size of the cell decreases further, and the nucleus becomes more densely packed with chromatin. It becomes kidney-shaped, and both azurophilic and secondary granules are observed. In the band cell stage, the cell size decreases even further, and the relative shape of the nucleus becomes more elongated and lobular [17].

Behind the differentiation of neutrophils from the hematopoietic stem cell, several molecules and factors have a crucial role. Granulocyte-colony stimulating factor is one of them. It has vital roles in the neutrophil life cycle, such as controlling the differentiation of neutrophils from the hematopoietic stem cells, the production and survival of neutrophils [29], and their release from the bone marrow into the blood [30,31]. Apart from this, chemokine receptors CXCR2 and CXCR4 control the decision of retention or release of the neutrophils from mouse bone marrow; CXCR4 (which is the receptor for the chemokine CXCL12) is mainly associated with the retention of neutrophils in the marrow and CXCR2 (which is the receptor of the CXCL1 and CXCL2) is related to the event of a neutrophil release from the bone marrow [32]. In humans, IL8 (CXCL8) is the major ligand for CXCR1 and CXCR2. On top of these, several other transcription factors, such as the CCAAT enhancer binding protein α gene [33] and PU.1 [34], are essential in the later stages of neutrophil differentiation.

## 3. Common Techniques and Methodologies Used to Study Neutrophil Biology

*Proteomics*: There are three main types of proteomics: expression proteomics, structural proteomics, and functional proteomics [35,36]. Expression proteomics quantitatively identifies and characterizes proteins in various sample groups (Control, treatment). Proteins are separated using two-dimensional gel electrophoresis, and then the separated proteins are identified using mass spectrometry (MALDI TOF MS, LC-ESI MS/MS) [37]. On the other hand, the primary objective of structural proteomics is to elucidate the structure and composition of proteins and study the protein–protein interactions [38]. In this proteomics, target proteins are first immunoprecipitated, then separated by gel electrophoresis. Finally, they are identified using mass spectrometry [37,39]. In contrast, functional proteomics is about studying the function of proteins [40], whereas mass spectrometry is widely used with other techniques [37].

Single-cell sequencing: This is a widely used state-of-the-art technique for studying neutrophils. The main benefit of this technique is that it can identify the expression of various genes and transcripts at a single-cell resolution [41]. For example, using single-cell RNA sequencing, it has been shown that the circulating population of neutrophils is heterogenous, contrary to the popular belief of one homogenous population of neutrophils [42]. On top of that, the data obtained from the single-cell sequencing of neutrophil are processed through sophisticated algorithms such as diffusion mapping, which lead to further advancement of understanding about the development of neutrophils [43].

Single-cell Metabolomics: Studying immune cells such as neutrophils with a short life span is challenging but essential. One of the ways to understand cellular metabolism is to study the whole metabolome, which is the collection of all metabolites at a particular time point [44]. The composition of metabolites is measured at the single-cell level to understand cellular metabolism processes in individual cells and populations. It has been recently used to discover the role of glutathione depletion in the spontaneous apoptosis of neutrophils [45].

Multiphoton Imaging: Another way to obtain real-time information about highly dynamic immune cells such as neutrophils is to use two-photon and three-photon imaging, which, unlike confocal microscopy, uses two photons for better penetration into the sample and less photodamage [46,47]. Using this technique, researchers have shown the extravasation of neutrophils. For example, micro emboli formation in sickle-cell disease and NET formation occurs in the lungs upon receiving inflammatory cues [48,49,50,51,52]. Similarly, another group of researchers utilized multiphoton imaging to study neutrophil-extracellular-trap formation in vivo [53].

## 4. Contribution of Neutrophils in Neurological Diseases

### 4.1. Ischemic Stroke

Ischemic stroke is defined by the occlusion of blood vessels in the brain due to thrombus formation. Within minutes, neutrophils in mouse blood can be detected at the ischemic site, guided by inflammatory cytokines such as Il-1, Il-6, CXCL1, and TNF and chemokines such as CCL-2, CCL-3, and CCL-5 [54,55]. In addition, damage-associated molecular patterns such as high mobility group box protein, heat shock protein, and DNA released by the death of the cells help in their recruitment and activation in mice and patients (Figure 2) [55,56]. Neutrophils then adhere to the endothelium and transmigrate inside the tissue space. The pro-inflammatory cytokines, proteases, and reactive oxygen species secreted by neutrophils exacerbate cerebral inflammation, cause blood–brain barrier breakdown, and recruit neutrophil reinforcement to the thrombus site [55]. Neutrophils can also exaggerate the pro-thrombogenic pathways by direct interaction with platelets, proteolytic cleavage of clotting factors, and releasing a neutrophil extracellular trap that binds and entraps platelets [57,58].

Further research has shown that the neutrophil population has functional heterogeneity. The pro-inflammatory ‘N1’ neutrophils have an anti-inflammatory ‘N2’ counterpart. The N2 neutrophils secrete TGFβ and IL-10 and have a neuroprotective and immune-suppressive function [59], but further research is needed to establish whether N1 and N2 neutrophils are ontogenically different. Recent studies in mice have shown that reducing the adhesion and neutrophil number using an antibody against the neutrophil protein Ly6G increased reperfusion by reducing the number of capillary obstructions in the capillaries [60,61]. Overall, the ischemic stroke outcome was less severe when neutrophils were depleted.

### 4.2. Parkinson’s Disease

Parkinson’s disease (PD) is a progressive neurodegenerative disorder that causes dopaminergic neuron loss in the brain, primarily in the substantia nigra, which has essential roles in controlling movement. Besides the accumulation of alpha-synuclein and loss of dopaminergic neurons in the substation nigra, PD encompasses both central and peripheral inflammation. Many studies uncover the involvement of immune cells, such as neutrophils, in PD [62]. In mice, one of the effector mechanisms of neutrophils is the use of large concentrations of nitric oxide, which is generated by nitric oxide synthase [63]. Patients with PD have elevated expression of neuronal nitric oxide synthase in their circulating neutrophils, indicating a role in increased oxidative stress in PD patients [64]. Nitric oxide reacts with superoxide anion and can give rise to peroxynitrite, which is shown to trigger the release of the neutrophil extracellular trap ex vivo, exacerbating inflammation [65]. Since neutrophils are one of the most abundant white blood cells in the blood, they are often utilized to assess peripheral inflammation. For example, leucine-rich repeat kinase two has essential roles in Parkinson’s disease. Its kinase activity is measured by the phosphorylation status of Rab10, which can be extracted from peripheral blood neutrophils [66]. The neutrophil-to-lymphocyte ratio is a way to estimate inflammation. Although its role in Parkinson’s is still under investigation, some studies show that the patients generally have a higher neutrophil-to-lymphocyte ratio than the healthy controls [62,67].

### 4.3. COVID-19

Severe acute respiratory syndrome coronavirus-2 causes COVID-19 disease, primarily damaging the lungs, and causes mild to moderate respiratory difficulty in many patients [68]. Inflammation, which occurs during the acute phase of infection, recruits neutrophils to the lungs, which then undergo degranulation with the secretion of NETs and pro-inflammatory cytokines. This results from animal models in an increase in inflammation and associated tissue damage [69]. Almost half the people with COVID-19 are now developing long-term symptoms [70]. Neutrophils, among the other leukocytes, play a vital role in the post-COVID-19 changes in the lungs and other organs, including the brain. Compared to control patients, people with post-COVID-19 interstitial lung changes have shown increased levels of pro-inflammatory cytokines such as TNF and IL-17C in the plasma and reported upregulation of proteins known for neutrophil chemotaxis such as CCL20 and CCL25 (Figure 2) [71]. This group also showed more circulating neutrophils (indicating peripheral inflammation), higher concentration of myeloperoxidase (peroxidase expressed predominantly by neutrophils), and increased NETosis compared to controls [71]. The upregulation of NETs during long COVID-19 increases the expression of fibrogenic mediators, which can lead to thrombosis in the lung, kidney, or other organs [71,72]. It is proposed that neutrophils specifically harm the microvasculature by causing endothelial cell swelling and small clots [73,74]. Patients with long COVID-19 also reported varying degrees of cognitive impairment, commonly called ‘brain fog.’ It is reported that SARS-CoV-2 can enter the central nervous system via dysregulation of the BBB [75,76]. Once inside the brain, it can constrict the capillaries and reduce the blood flow via binding ACE2 on the pericytes [77]. SARS-CoV-2 also increases the viscosity of the blood, which causes the reduced blood flow in the cerebral capillaries [78]. As seen in other dementia or stroke models, this reduction in the blood flow might be one of the mechanisms behind cognitive impairment in the long COVID-19.

### 4.4. Huntington’s Disease

Huntington’s disease (HD) is a rare neurodegenerative disease that causes neurodegeneration due to a mutation in the Huntington gene [79]. For a long time, the role of the immune system in HD has remained underappreciated, but the recent focus on this area has started to uncover some fascinating insights. For example, compared to the healthy controls, the plasma of HD patients has been shown to contain increased IL-6, MMP-9, VEGF, TGF-β1, and decreased IL-18 levels [80]. Interestingly, Il-6 deficiency in a mouse model has been shown to exacerbate the HD symptoms and thus may point towards the importance of optimal IL-6 in the body [81]. In HD cases, it is thought that there is only a marginal influx of peripheral immune cells, including neutrophils [79]. In another study, however, the effect on neutrophils went in the other direction. Using BACHD mice, a well-characterized mouse model of HD, the authors observed that the BACHD mice had 60% fewer infiltrating neutrophils in the peritoneum than wild-type mice [81]. More data need to be reported on the effect of neutrophils on mouse models and patients with HD.

### 4.5. Amyotrophic Lateral Sclerosis

Amyotrophic lateral sclerosis (ALS) is a neurodegenerative disease caused by neurogenic amyotrophy and degeneration of upper and lower motor neurons [82]. Dysregulated inflammatory pathways are prevalent in patients with amyotrophic lateral sclerosis, characterized by increased levels of pro-inflammatory cytokines and subsequent immune cell infiltration into the CNS, such as neutrophils [83]. Relatively high levels of inflammatory cytokines, such as IFN-β, TNF-α, IL-6, and IL-8, were identified in the plasma of ALS participants. These pro-inflammatory cytokines have an imperative role in ALS mouse models in neutrophil activity and trafficking [83,84,85]. High levels of IL-6 and sIL-6R have been demonstrated in several chronic inflammatory and autoimmune diseases [86,87]. A study has shown the effect of systemic IL-6-mediated inflammation on endothelial cell (EC) death damaging the CNS barrier in ALS individuals (Figure 2) [88]. In patients, IL-6R can be released by proteolytic shedding from neutrophils or by secretion from monocytes of an alternatively spliced messenger RNA (mRNA) species as a soluble form (sIL-6R). Increased shedding of sIL-6R promotes IL-6/sIL-6R complex formation in the blood [89,90]. This complex activates the trans-signaling pathway upon binding to glycoprotein 130 on the target cell membrane, activating JAK and other signal transduction molecules, including MAPK, ERK, P13K, and STAT. This results in a pro-inflammatory response in the endothelial cells and a de novo synthesis of monocyte-attracting chemokines and vascular cell adhesion molecules, leading to the extravasation of inflammatory cells into the CNS [88]. It is believed that such a swamp of cells will cause EC degeneration, the principal cause of damage to the brain–CNS barrier in ALS.

Apart from the role of inflammatory cytokines, other immune factors are associated with ALS that can enhance the activity, trafficking, and survival of neutrophils, including Leukotriene B4, platelet-activating factor, and C5a [91]. Several studies have shown a correlation between the increased percentage of neutrophils within the total leukocyte population and disease progression [92,93,94]. Accordingly, the increased neutrophil-to-monocyte ratio in patients with ALS suggests progressive alterations in peripheral myeloid populations, which may contribute to functional changes associated with the disease [92]. In the skeletal muscle, neutrophils are recognized for their role in mediating myofiber damage and atrophy. It was found that ALS patients have significant infiltrations of mast cells and phagocytic neutrophils into their skeletal muscles. These cells orchestrate interactions with each other, myofibers, and motor endplates, leading to neuromuscular junction denervation and muscular atrophy. In mouse models, an influx of endomysial elastase-expressing neutrophils may cause such atrophy in muscles, which is reportedly upregulated in muscular dystrophy, impairing myogenesis [95,96]. The data suggest that neutrophil hyperactivation with a high potential for cytotoxicity is involved in neutrophil-mediated inflammation due to NET formation, indicating that the uncontrolled activation of neutrophils likely contributes to muscular pathology ALS.

Studies have shown a link between immune cell features, such as population frequency and expression markers, and disease features. Accordingly, CD16 (FcgRIIIb) expression on mouse neutrophils is associated with disease severity and disease progression rate. As neutrophils age in culture, they lose expression of CD16 in parallel with declining functionality, increasing phagocytosis and oxidative stress [96,97]. This suggests the possibility of reactive oxygen species production during chronic neutrophil activation that exacerbates motor neuron degeneration [91]. A study by Murdock et al. showed that the immune system affects ALS patients differently depending on their gender. Using two separate survival models, the study examined the potential role of neutrophils in ALS patients and determined whether sex affects immunity in those individuals [98]. First, they showed the association of increasing peripheral neutrophils with increased mortality in ALS [99,100]. Second, they revealed that the role of neutrophils is complicated by sex, as low peripheral neutrophil levels were associated with more prolonged median survival in females. Low neutrophil levels, however, did not correlate with increased survival in male participants. This discrepancy could be explained by various mechanisms, including the direct effects of sex hormones on neutrophil activity due to the immunomodulatory effects of male and female hormones and the alternation of the immune environment by sex within the CNS [92,94]. The data suggest a profound impact of neutrophils in the disease progression, and sex-specific neutrophil responses could contribute to sex differences seen in ALS patients. Much more work is needed to untangle the role of neutrophils and their sex-specific roles.

### 4.6. Multiple Sclerosis

Multiple sclerosis (MS) is a chronic inflammatory disorder of the central nervous system (CNS), characterized by the deterioration of the myelin sheath that protects the nerve fibers. Neutrophil infiltration into the CNS, followed by cytokines secretion such as TNF-α, IL-6, IL-12, IL-1β, and IFN-γ, triggers the inflammatory cascade, causing BBB injury in MS patients [101].

IL-17, known to be produced by Th17 cells and neutrophils, is also reported to be upregulated in the brain of experimental autoimmune encephalomyelitis (EAE) mice during the early stages of the disease. IL-17 is known to disrupt the BBB, which facilitates the migration of inflammatory cells into the brain and also affects the production of other inflammatory chemokines and cytokines associated with EAE, including CXCL1 and CXCL2 [102,103]. Observations suggest neutrophils may damage neurons directly through CXCR2 signaling, a key regulator of neutrophil neurotoxicity. A study found that neutrophils isolated from control EAE mice induced severe neuronal cell death via CXCR2-mediated ROS generation in neutrophils. Neutrophil-specific CXCR2 deletion is sufficient to rescue this effect by preventing ROS production. We can conclude that MS’s pathogenesis requires CXCR2 signaling [104,105].

Increased neutrophil numbers are primarily attributed to a decrease in spontaneous apoptosis. The existence of a pro-inflammatory environment appears to modify neutrophil lifespan in patients with relapsing-remitting MS (RRMS), as it was shown that RRMS neutrophils have greater apoptosis resistance than neutrophils from healthy controls (HC) [106], stating that an inflammatory environment can influence neutrophil survival and apoptosis. Such an environment can result from inflammatory priming agents such as IL-1β, IL-6, IFN-y, G-CSF, GM-CSF, or IL-8, which can delay neutrophil apoptosis. In addition, elevated levels of TLR2, CD43, FPR1, and CXCR1 in RRMS patients’ neutrophils further suggest that the chronic inflammatory milieu primes neutrophils in RRMS [107]. Primed neutrophils from RRMS patients showed enhanced effector mechanisms, including degranulation, oxidative burst, and release of NETs [108]. Increased levels of NETs are linked to pathophysiological conditions, which are believed to be induced by inflammatory mediators, including IL-8. MS patients show higher levels of IL-8 [109], which also prolongs neutrophil survival [110]. Interestingly, a study found that the subset of RRMS patients with high NETs in the serum was significantly enriched in male patients compared to females, suggesting that NETosis may be gender-specific in this disease and could be involved in certain aspects of MS pathogenesis, such as the opening of the BBB [111].

### 4.7. Autism

Autism spectrum disorder (ASD) is a pediatric heterogeneous neuropsychiatric disorder characterized by social and communication deficits, language impairment, and ritualistic or repetitive behaviors [112]. Dysregulations in innate and adaptive immune cells have been implicated in the pathogenesis of ASD [113]. Thereby, oxidants and pro-inflammatory cytokines generate important effects.

Past investigations found high levels of pro-inflammatory cytokines, including IL-6 and IL-17A, in ASD patients and autistic mice [114,115]. In ASD subjects, IL-17A produced by immune cells such as Th17 and monocytes/macrophages cells can activate neutrophils. Primed neutrophils might worsen overall inflammation, indicating they play a crucial role in developing peripheral inflammation. A study has found the upregulation of IL-17A/IL-17R in neutrophils of ASD patients, whose interaction is required in oxidative stress regulation in neutrophils. Activation of IL-17A/IL-17R in neutrophils correlates with elevated NOX2/ROS signaling [113].

Excessive oxidative stress due to the activation of NFκB and iNOS is also reported in autistic subjects and BTBR T + Itpr3tf/J (BTBR) mice. This study showed that sulforaphane, an Nrf2 activation mediator, inhibited NFκB-iNOS signaling in neutrophils and cerebellum [116], given the importance of Nrf2 in controlling immune system response and cellular oxidative stress. It was suggested that sulforaphane would attenuate the effects of oxidative stress on neutrophils and brain tissue in autism via activating Nrf2 in microglial cells and macrophages.

ASD neutrophils may also cause increased oxidative stress in response to environmental pollutants such as i-2-ethylhexyl phthalate (DEHP). Neutrophils in such patients are unable to mount an antioxidant response, which was shown by lower expression levels of Nrf2 [117].

Epithelial cell-derived neutrophil-activating peptide-78 (ENA-78/CXCL5) is a CXC chemokine that attracts and activates neutrophils [118]. Accompanied by IL-8, it can induce neutrophil chemotaxis and activation by increasing intracellular calcium levels and elastase release [119], stating that the dysfunction of the immune system can act as a crucial factor in the development of autism.

Highly reactive neutrophil cathepsin B can initiate leukocyte–endothelial cell adhesion by promoting leukocyte Mac-1 activation and its interaction with endothelial ICAM-1. Accordingly, Mac-1 in neutrophils and ICAM-1 in endothelial cells is found to be significantly higher in autistic mice, underlying the role of neutrophil cathepsin B in inducing neurovascular inflammation during autism [120].

### 4.8. Down Syndrome

Down syndrome (DS) is one of the most common genetic disorders caused by complete or partial triplication of chromosome 21 [101]. Individuals with DS have a higher risk of Alzheimer’s disease because patients have an extra copy of the amyloid precursor protein (APP) gene located on chromosome 21, which is trivalent in this population [121]. Due to mutations in APP and genetic risk genes, such as PSEN1 and PSEN2 which alter amyloid concentrations, patients with DS are more prone to develop Alzheimer’s disease at an earlier age (30–60 years) than those with late-onset Alzheimer’s disease (≥65 years) [122].

It is generally accepted that patients with DS are susceptible to developing infections and chronic inflammatory conditions [122,123], which might also play a critical role in the onset and progression of dementia in these individuals. A meta-analysis study found high expression levels of Il-1β, TNF-α, IFN-γ, and IL-6 cytokines in DS patients compared to healthy control [124]. These cytokines stimulate neutrophil activation and their trafficking into the CNS, underlining the importance of primed neutrophils in neuroinflammation in DS patients. Elevated levels of inflammation can lead to Toll-like receptor (TLR) signaling dysregulation. A study reported the increased expression levels of TLR2 on neutrophils and altered gene expression of key regulators proteins involved in its signal propagation [125]. It is possible that excessive cytokine levels in DS may reflect abnormal signaling of TLR pathways. Since the end point of these pathways results in the release of inflammatory mediators, DS subjects may display a hyperresponsive immune reaction [124]. Furthermore, TLR2 and TLR4 regulate neutrophil function in DS patients, including apoptosis, adhesion, and activating pro-inflammatory signaling pathways, including the NF-κB pathway. Stimulation of the NF-κB pathway induces transcriptional machinery of pro-inflammatory factors, such as iNOS, NOX2, IL-6, and other chemokines/cytokines [10,113]. It has also been found that platelet TLR4 activation can lead to robust neutrophil activation and NET formation, which can cause BBB damage in DS subjects. The same symptoms were observed in Alzheimer’s disease when NETs develop intravascularly and intraparenchymally [10,14,126]. These observations raise the possibility that the same pathologic mechanisms might contribute to the progression of both diseases.

Although many of the inflammatory pathways described for AD would apply directly to DS, there are some particular inflammatory genes on chromosome 21 that affect inflammatory responses in the DS brain, including a highly expressed CXADR gene, which functions as an adhesion molecule and is associated with endothelial tight junctions [127,128]. It has been shown that CXADR not only can induce stress-activated mitogen-activated protein kinase (MAPK) pathways in the heart leading to increased production of IFNγ, IL-12, IL-1β, TNFα, and IL-6 but also has a significant role in BBB permeability facilitating trans-endothelial migration of neutrophils [127]. The increased migration of neutrophils damages the tight junction and increases the permeability of the BBB. This damage occurs when neutrophil elastase (NE) is released, disrupting cadherin–cadherin binding, thus increasing BBB permeability [129,130]. We can conclude that altered expression of the CXADR gene on the endothelial cells of the cerebral vasculature in DS subjects can affect inflammatory cells’ infiltration into the brain, such as neutrophils, and influence the inflammatory response.

DS patients are generally believed to be exposed to excessive amounts of oxidative stress due to dosage imbalances of the cytoplasmic enzyme Cu^2+^/Zn^2+^ superoxide dismutase (SOD-1) gene, which is also found in Chr 21 [131,132]. SOD plays a fundamental role in regulating reactive oxygen species (ROS) levels, a major contributor to oxidative stress, neuronal death, and disease progression [133]. Chronic neutrophil activation may exacerbate motor neuron degeneration in DS by producing reactive oxygen species. Indeed, neutrophils have been implicated in other neurodegenerative diseases such as AD [134] and produce significantly higher levels of reactive oxygen species in these patients [135], which is plausible since the functionally deleterious role of neutrophils in AD could be mirrored in DS. We can therefore suggest inhibiting NADPH oxidase enzymes (NOX) as a potential target to reduce ROS, leading to decreased capillary stalling and increased CBF [136].

### 4.9. Frontotemporal Dementia (FTD)

FTD is the second most common form of dementia and is a collection of neurodegenerative disorders affecting mainly the frontal and temporal lobes of the brain. Its symptom can vary widely between persons, but some include behavioral changes such as a lack of empathy, increased inappropriate social behavior, speech and language problems, and motor disorders [137]. Neuroinflammation, macrophage infiltration, and microglial activation are reported in some patients with FTD [138,139]. Although microglia are thought to be one of the critical immune cells in the progression of FTD [140], recent data indicate the involvement of other immune cells in the disease progression. Employing a mouse model with frontotemporal lobular degeneration, the authors observed the differential expression of genes associated with neutrophils and monocytes [141]. Moreover, the presence of the neutrophilic protein, 37 kDa cationic antimicrobial protein (CAP37), is detected in the brain of a patient with FTD, indicating the involvement of neutrophils in this disease [21].

Another critically important protein in FTD is progranulin, whose mutation is a major cause of the onset and progression of the disease [142]. Progranulin involves multiple pathways, from wound healing after injury to inflammation [143]. Single-nucleus RNA sequencing of FTD patients’ cerebral cortex has shown profound neurovascular dysfunction [144]. Interestingly, there is also a dynamic interaction between neutrophils and progranulin; neutrophils release serine protease, and elastase can cleave progranulin into smaller fragments of granulin peptides [145]. At least, one of the granulin fragments, granulin-B, has been shown to elicit an inflammatory response [142]. In contrast, another neutrophil-secreted protein, leukocyte protease inhibitor-1 (SLP-1), can inhibit the generation of granulin fragments from progranulin [146]. Therefore, understanding the role of innate immune cells such as neutrophils in the progression or resolution of FTD is crucial and might lead to the development of therapeutic interventions.

## 5. Contribution of Neutrophils in Alzheimer’s Disease (AD)

An increasing number of studies provide evidence indicating that inflammation and the innate immune system play a pivotal role in the development of AD [147]. Therefore, neutrophil activation and alteration may contribute to increased vascular inflammation associated with AD [148]. A study conducted with a transgenic mouse model of AD revealed that neutrophil depletion significantly reduced Alzheimer-like neuropathology and improved memory in mice with cognitive deficits [14]. One of the pathological hallmarks of Alzheimer’s disease is the generation of senile plaques that make up beta-amyloid peptides (Aβ) [149]. Aβ plays an essential role in recruiting neutrophils to inflamed tissues by triggering a shift in lymphocyte function-associated antigen-1 (LFA-1) from a low to high-affinity state, resulting in increased LFA1-dependent adhesion of neutrophils in AD mouse models [14]. High-affinity LFA-1 may also signal the arrest of neutrophils within the parenchyma, causing neutrophil accumulation and leading to the widespread neutrophil-dependent central nervous system (CNS) damage [14]. The data suggest an interaction between the innate immune system and senile plaques linking the periphery with the CNS [150]. Furthermore, the adhesion molecule, Mac-1 (CD11b/CD18), is expressed at higher levels in sporadic AD patients than in control patients. This increase may be correlated with disease severity and the progression of dementia [151]. It should also be noted that the α subunit of Mac-1, CD11b, is thought to be expressed exclusively on leukocytes, where it interacts with the adhesion molecule-1 (ICAM-1, CD54) and other endothelial surface ligands [152]. In this regard, the significant elevation in blood neutrophils, Mac-1, in AD patients suggests a state of neutrophil activation. This activation is likely a consequence of increased levels of TNF-α, IL-6, ICAM-1, CRP, and other inflammatory-immune markers observed in AD patients’ serum (Figure 3) [151,153,154].

Neutrophil-associated inflammatory proteins may affect memory and executive function in patients with AD. Lipocalin-2 (LCN2), a pro-inflammatory molecule secreted into the peripheral circulation by neutrophils, is believed to cause cognitive decline [148,155]. Accordingly, several studies have reported the upregulation of LCN2 in AD mouse models and patients’ serum and brain tissue [156,157,158]. How LCN2 contributes to cognitive decline in AD patients is not fully understood. LNC2 may affect neutrophil adhesion by altering the expression of the neutrophil adhesion factors CD62L (L-selectin), Mac-1 (CD11b/CD18.), and CD51/CD61 (αvβ3) [159], and the increased adhesion causes an over activation of the inflammatory cascade. The overactivation, in turn, may reduce proper cellular function contributing to neuronal dysfunction. It is unclear how LCN2 contributes to cognitive decline in AD patients. It may affect neutrophil adhesion by altering the expression of neutrophil adhesion factors CD62L (L-selectin), Mac-1 (CD11b/CD18.), and CD51/CD61 (αvβ3). It is also possible that LCN2 induces chemokine production and enhances neutrophil trafficking into the brain. In addition to LCN2, four different neutrophil adhesive and activation promoter proteins have been associated with executive function deficits in mild AD, including MPO, IL-8, TNF -α, and MIP-1β (CCL4) [148]. We can therefore conclude that neutrophil-associated inflammatory proteins and their related pathways potentially contribute to AD progression and severity.

The tight cohesion of leukocytes causes capillary stalling to the cortical capillary endothelium. This adhesion contributes to cerebral blood flow (CBF) reduction in mouse models of AD. Studies have shown the contribution of pro-inflammatory mediators such as IL-1β, IL-6, and TNF-α increased vascular inflammation and likely increased capillary stalling [160]. Elevated levels of inflammatory cytokines can enhance neutrophil infiltration, which may also contribute to a decrease in CBF. In this regard, Cruz Hernández et al. investigated the role of neutrophil adhesion in capillary segments, which exacerbates AD symptoms due to CBF reduction. By administrating a fluorescently tagged antibody against the neutrophil marker Ly6G, they could detect immediate improvement in CBF in amyloid precursor proteins (APP) mice [134]. In patients and mouse models of AD, elevated NADPH-oxidase 2 (NOX2) increases oxidative stress in the microvasculature, and oxidative stress contributes to capillary stalling and vascular inflammation. A recent preprint has demonstrated in an AD mouse model, that inhibiting NOX2 improves short-term memory [135,161]. These improvements were associated with increases in CBF by reducing capillary stalling in the APP/PS1 mouse model [135]. Although neutrophils impact AD progression and severity, very little is known about the mechanism affecting neutrophil changes.

Neutrophils tend to act in different forms when pathogens invade. These changes include ROS production causing an oxidative burst, phagocytosis, and degranulation. Neutrophils have elevated amounts of NADPH-oxidase, which has the essential function of generating reactive oxygen species [162]. Elevated levels of ROS lead to neutrophil hyperactivation and increase NET formation. As neutrophils are a crucial medium for inflammation-activated vascular and tissue damage, the inappropriate activation of neutrophils may cause oxidative stress and magnifies inflammatory responses. Neutrophil activation can also trigger apoptosis and hamper their ability to function full-fledged against the impulse [134]. The experimental analysis noted that the flow of aged neutrophils increases ROS production, which is directly related to Aβ deposition in patients [163,164].

## 6. Conclusions and Future Perspectives

It has become evident that neutrophils are not only important as first responders after infection or injury following a static response. Neutrophils can react dynamically and highly sophisticatedly to orchestrate a response fighting the threat and attract other immune cells. In aging and neurological diseases, neutrophils appear more activated, likely driving a vicious cycle, further increasing the low-level chronic vascular inflammation associated with neurological diseases. These reactive neutrophils will likely release pro-inflammatory cytokines and chemokines, forming more NETs in the vessels and the parenchyma. Moreover, the inflammasome was shown to be more active in neurological diseases such as Alzheimer’s, driving further NET formation [165,166]. Altogether, these factors drive an inflammatory environment in the periphery and the brain even further.

It is imminent, despite their short lifespan, that neutrophils are contributing to the increased inflammation associated with neurological disease development, so that inhibiting reactive neutrophils could contribute to slower disease progression. Therefore, a better understanding of neutrophil behaviors in neurological diseases will be crucial to better understanding the transition to more reactive neutrophils. The question of neutrophil heterogeneity is still very much open and needs further investigation using singe-cell and other approaches. Overall, due to the nature of neutrophils, in vivo, studies are critical to investigate neutrophils in the tissue and their interaction with the vasculature or sites of injury.

Neutrophils are significantly different between rodent models and humans, and this needs to be considered when interpreting results. Furthermore, sterile housing conditions also significantly impact neutrophil maturation and the immune system in model organisms [167,168]. For example, wild mice can be used or co-housed with pet-shop mice to make the immune system more mature [169]. Another example to address this problem is the recently humanized neutrophil mouse model [170]. More studies are needed to overcome these shortcomings.

There are many open questions which mean more research in model organisms, and patient is needed to understand the specific contributions of neutrophils in neurological diseases. However, neutrophils eventually drive inflammation in the vasculature and contribute to aging and age-related disease.

## Figures and Tables

**Figure 2 biomolecules-13-00743-f002:**
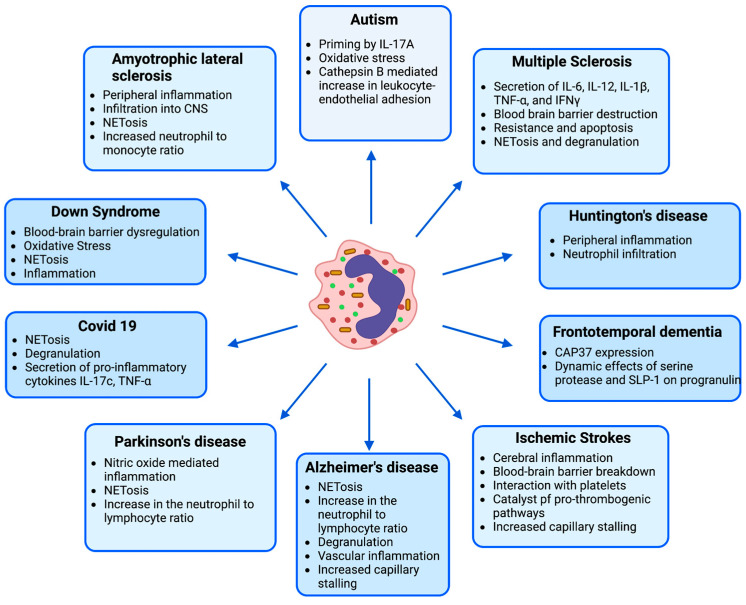
An overview of neutrophils’ functions in various neurological diseases.

**Figure 3 biomolecules-13-00743-f003:**
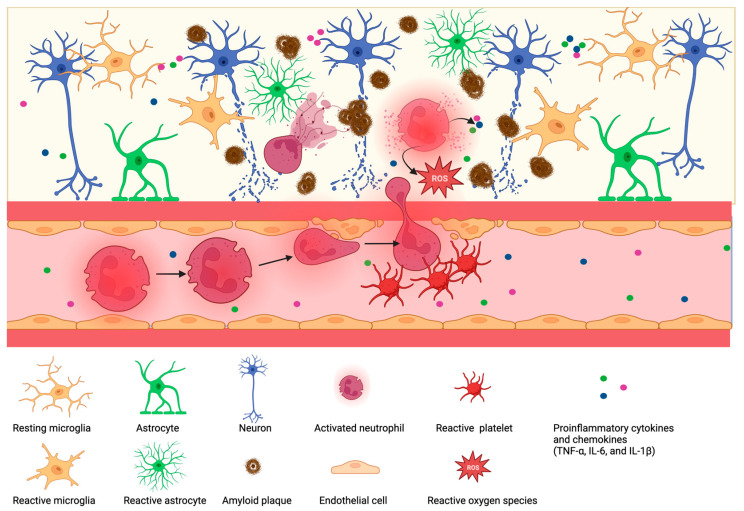
Diagrammatic representation of cellular events in the Alzheimer’s vasculature. Increased deposition of amyloid plaques in the brain leads to inflammatory processes; reactive microglia and astrocytes release pro-inflammatory cytokines IL-1β, TNF-α, and IL-6 are released to the bloodstream and attract and activate innate immune cells such as neutrophils and platelets. Peripheral inflammation contributes to reactive neutrophils and platelets. Those, in turn, are more prone to react with blood vessels, likely at sites of reduced blood–brain barrier permeability. Neutrophils are now more likely to stall blood vessels and degranulation inside the vessels. Furthermore, neutrophils interact with reactive platelets, transmigrate into the brain parenchyma, undergo degranulation and NETosis, and secrete reactive oxygen species and pro-inflammatory chemokines and cytokines. This vicious cycle contributes to low-level chronic inflammation in patients with Alzheimer’s.

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
