# Peer review of "A Brief Overview of Neutrophils in Neurological Diseases"

_biomolecules, 2023, doi:10.3390/biom13050743_

Round 1

Reviewer 1 Report

In the review article by Chakraborty et al, authors have provided a comprehensive overview of what is known about the potential role of neutrophil-mediated immune response in neurological disorders. It will be an important source of information for readers interested in learning how neutrophil contribute to tissue injury in auto-immune or auto-inflammatory disorders both in and outside of the field of neurobiology. However, there are several limitations with the current version of the manuscript that require authors attention. Also, the manuscript is full of grammatical mistakes and missing punctuation all over the manuscript. Authors should address following concerns:

1.     Oxidative burst without NETs generation is also one of the effector functions of neutrophils. On line 9 of page 1 (abstract), authors should write following “the release of proinflammatory cytokines and chemokines, oxidative burst and the formation of neutrophil extracellular traps”.

2.     P-selectin glycoprotein ligand-1 (PSGL-1) on neutrophils is the constitutive ligand for endothelial P-selectin. Also, the statements in this paragraph are not accurate. On page 1, line 27-30, authors should write following “P-selectin on endothelial cells, which interacts with the sialo-mucin ligand P-selectin-glycoprotein-ligand-1 (PSGL-1), constitutively expressed on the surface of neutrophils. This interaction slows down the speed of free-flowing neutrophils in the blood vessel by mediating neutrophil rolling (stop and go movement) along the endothelium”.

3.     ICAM-1 and -2 are not integrins. These are ligands for CD18 integrins CD11a/CD18 (LFA-1) and CD11b/CD18 (Mac-1) expressed on neutrophils. On page 1, line 29-31, authors should write following “Following the step of neutrophil rolling, a high affinity interaction of CD18 integrins CD11a/CD18 (LFA-1) and CD11b/CD18 (Mac-1) expressed on neutrophils with inter-cellular-adhesion-molecule-1 or -2 (ICAM-1/ICAM-2) expressed on endothelium mediates the firm arrest (complete stop) of neutrophils on the vascular endothelium”.

4.     On page 1, line 41-42, authors should also mention a fourth mechanism “Neutrophils also undergo oxidative burst”.

5.     Page 2, line 58, why would a “Results” heading be used in a review article. Please remove.

6.     In Figure 1, authors are describing only murine neutrophil markers. Authors should also include human neutrophil markers in the figure 1. Human neutrophils are CD16+/CD66b+ dual positive and may express CD14 at low levels.

7.     Line 88, page 3. It should be “Granulocyte-colony stimulating factor is one of them”. The grammar and sentence structure is inappropriate at several places throughout the manuscript. The manuscript needs text editing.

8.     Page 3, line 93-95 “CXCR2 (which is the receptor of the CXCL1 and CXCL2) is related to the event of a neutrophil release from the bone marrow [29].” In humans, IL8 (CXCL8) is the major ligand for CXCR1 and CXCR2. Authors should mention it here. This problem is present all over the manuscript. At several places, authors are describing without mentioning whether these findings were made in human or mice neutrophils. This needs to be fixed all over the manuscript otherwise it will lead to misinformation among readers.  

9.     Page 4, line 118, it should be “various genes and transcripts at a single-cell resolution [38].” Please proof read for grammatical errors all over the manuscript. Same issue on line 128, page 4. It should be “understood at a single-cell resolution”.

10.  Page 4, line 137. Authors need to cite more relevant intravital; studies from other investigators. Please cite PMID: 21151136; PMID: 35737916; PMID: 33226733; PMID: 28097236.

11.  Page 4, line 141 should be “Ischemic stroke is defined by occlusion of blood vessels”.

12.  Page 4, line 143. Authors seem to describing murine studies. IL8 is not present in mice. As mentioned in previous concern, it is hard to understand at several places in the manuscript whether authors are referring to human or mice studies. This is a major concern, which needs to addressed. Also, several places mention interleukin as Il instead of IL. Authors need to be consistent.

13.  Line 146, page 4, neutrophils do not “attach” but they “adhere” to endothelium.

14.  Page 4, line 150-152. Neutrophils do not cleave coagulation factors themselves but they indirectly facilitate the cleavage of coagulant factors. Please correct it.

15.  Line 157-160 on page 5. It is very hard to understand whether these findings were made in mice or humans. This issue is everywhere in the manuscript.

16.  Page 6, line 233-236. Very hard to understand this sentence “A study has shown that unstimulated neutrophils in healthy donors' peripheral blood express increased levels of IL-6 [79]. IL-6 is binding to the constant portion of immunoglobulin G (IgG) surface Fc receptors [80], leading to the activation of neutrophils”. Does it mean neutrophils from ALS patients? If yes then it needs to be rewritten in a way it can be easily understood.

17.  Page 7, line 264-267. CD16 is expressed only on human neutrophils but this sentence does not clarify it.

18.  Page 7, line 267. “With aging, neutrophils lose expression of CD16…”. What does aging means here? Are authors referring to neutrophils from aged humans or they are referring to aged neutrophil?

19.  Page 7, line 289-290, please correct the grammar.

20.  Page 8, line 291-293. IL17 also promotes neutrophil production and release from the bone marrow into the peripheral blood. Citation: PMID 26252407. This needs to be included.

21.  Page 8, Line 300-301 shows an incomplete sentence.

22.  Page 8, Line 301-302 “In EAE, CXCL16 and its receptor CXCR6 are detected at high expression levels,”. They are expressed where? Which cells?

23.  Page 8, line 312. Replace “reinforce” with “further suggest”.

24.  Page 8, line 318. Citation is missing.

25.  Page 9, line 352-353. Please replace with following “Highly activated neutrophil cathepsin B can initiate leukocyte-endothelial cell adhesion, by promoting leukocyte Mac-1 activation and its interaction with endothelial ICAM-1”.

26.  Page 9, line 353. Replace “Accordingly, CD11b…” with “Accordingly, Mac-1…”.

27.  Page 10, line 414. Citation is missing.

28.  Page 11, line 457-458 “Mac-1 (CD11b/CD18), is expressed at higher levels in sporadic AD patients….”. This sentence is incomplete. Mac-1 is highly expressed where or which cells? I am assuming neutrophils but it needs to be mentioned.

29.  Page 11, line 462-463 is very hard to understand. It needs to be written in an intelligible way.

30.  Page 11, line 472. Mac-1 is not CD11b. It is CD11b/CD18.  Similarly, CD53 is not avB3. CD53 is only the av chain. Please use correct nomenclature. Also, avb3 is not expressed on neutrophils or at least it does not promote neutrophil adhesion to endothelium. If authors believe it does then it has to be supported by a citation.

31.  Page 12. Line 502-505. The two sentences are hard to understand. Seems some information is missing.

32.  Page 12, line 516-517. “neutrophils are contributing to the increased inflammation associated and..”. Associated to what? This is an incomplete sentence.

33.  Page 12, line 532, replace “patience” with “Patient”.

Author Response

We apologize for the number of misspellings and grammatical problems. We thank the reviewers for their helpful comments and have addressed all suggested changes. 

Reviewer 2 Report

The manuscript contains a lot of knowledge about the role of neutrophils in nneurological diseases. The presented data will certainly contribute to the systematization of the current knowledge on the mechanisms of neutrophils action in selected diseases.

However, some changes and numerous fixes are necessary:

a/ more figures would allow a better understanding of the issues discussed

b/ Fig.2. adds nothing new to the knowledge about neutrophils, it is unnecessary

c/ there are factual errors in the manuscript, e.g.:

- ICAM-1/ICAM-2 are not integrins

- I have not found information in the cited literature regarding the increase in the level of IL-6 in neutrophils of healthy donors. What may be the cause of the increase of IL-6 in healthy donors?

There are many spelling errors in the text:

- Il-1 instead of IL-1, etc.

- IFN-y instead IFN-gamma

- Diapedesis instead diapedesis

In conclusion, other directions of research on neutrophils in neurological diseases, such as the formation of the NLRP3 inflammasome, should be indicated.

Author Response

(The authors gave the same response as above.)

Round 2

Reviewer 1 Report

 CD51 is just the alpha v chain of alpha v beta 3 integrin. The correct nomenclature for the hetromeric complex alpha v beta 3 is CD51/CD61 but not just CD51. CD61 is the beta 3 chain. Please address this on page 11. 

Author Response

Thank you for pointing this out. Is is correct that the Integrin alpha v beta 3 must be changed to CD51/CD61. We have corrected this in the current version